# Identifying Intention-Based Factors Influencing Consumers' Willingness to Pay for Electric Vehicles: A Sustainable Consumption Paradigm

**ShiYong Zheng** [1,2,3] , **Hua Liu** [1] , **Weili Guan** [2,*] , **Yuping Yang** [1] , **JiaYing Li** [1] , **Shah Fahad** [3] and **Biqing Li** [1]

1   School of Business, Guilin University of Electronic Technology, Guilin 541004, China
2   College of Digital Economics, Nanning University, Nanning 530021, China
3   School of Management, Hainan University, Haikou 570208, China
*   Correspondence: wlguan@126.com

**Abstract:** In an effort to reduce environmental pollution and energy consumption, the Chinese government strongly promotes the usage of electric vehicles. However, studies focusing on assessing consumers' willingness to pay (WTP) for electric vehicles (EVs) are limited in the country. This research aims to address this research gap by examining influencing factors of consumers' WTP for EVs in the Chinese perspective. Combined with the existing consumers' intention factors, the current study further contributed by augmenting the theoretical framework of the Theory of Planned Behavior by including three new dimensions (performance expectancy, information loaded, and perceived risk) to comprehensively analyze the willingness of Chinese consumers. Analysis is performed on survey data from 498 consumers using EVs in Beijing, China. To evaluate formulated hypotheses, structural equation modeling approach is employed. Empirical findings reveal that environmental knowledge and performance expectancy positively and significantly influence behavioral intention. In contrast, overloaded information has a negative impact on behavioral intention. Moreover, subjective norms are significantly and positively related to behavioral intention. The research outcomes further disclose that perceived risk is positively and significantly related to behavioral intention. Finally, behavioral intention has a significant and positive association with WTP for EVs. The study contributes to the literature on sustainable consumption behavior and provides academics and practitioners with essential future directions.

**Keywords:** environmental knowledge; sustainable consumption; performance expectancy; information overloaded; behavioral intention; willingness to pay for electric vehicles



## 1. Introduction

Reducing carbon emissions, driven by climate change mitigation motives, continues to be a prime environmental challenge faced by global economies [1–3]. In order to achieve the United Nations (UN) Sustainable Development Goals, transportation decarbonization is a critical aspect, and electric mobility plays a critical role in this area [4]. The electric vehicle industry (EV) has grown rapidly in the past decade due to government policies and technological advances [5,6]. During the past four years, selling electric vehicles has increased by an average of 60% each year [7]. Approximately 90% of global EV sales take place in China, Europe, and the United States. Globally, there are over 8 million electric vehicles, including both passenger cars and medium/heavy trucks, over half of which are in China [8]. In 2019, EV sales in the global market slowed down, particularly during the second half of the year, due to the contraction of the car market and reductions in EV subsidies across the major markets [9]. The percentage of EVs in the global car market is relatively small (2.6%), making it difficult to achieve mobility electrification in the near future. There was an increase of 256.56 million tons in China's gasoline and diesel consumption between 2012 and 2020 [10].

The higher expense of EVs, the lesser availability of charging infrastructure, and the extended charging time prevent consumers from purchasing electric vehicles [11]. The Chinese government has introduced several fiscal and non-fiscal policies to facilitate the acceptance of EVs. There are several forms of fiscal policy in place, including rebates on purchases, tax exemptions for purchases, infrastructure construction subsidies, and subsidies for electricity prices. There are primarily non-financial policies, such as free public charging at public charging stations and exemptions from road tolls [12]. It is important to note that EVs are sold poorly in the private sector, where the Chinese government is more interested in seeing EVs become a great success. The consumer's preference for EVs is the most critical factor in the private sector [13]. Due to this, it is essential to conduct a comprehensive study of the key factors affecting consumers' adoption of electric vehicles.

Several studies have investigated consumers' perceptions of environmentally friendly products [14]; however, very limited studies specifically focused on green vehicles, and most were contradictory. Despite having higher levels of ENK, consumers in emerging countries generally show a relatively low level of environmental awareness, which suggests further research is needed in this area. According to Said et al., participants are aware of a few local environmental concerns, but do not understand how to do sustainable consumption practices [15]. This study aims to fill the knowledge gap regarding emerging countries' consumers' BIs toward green vehicles. It is expected that the demand for green vehicles will increase, and very little is known about consumers' attitudes towards green vehicles, especially young consumers. When developing marketing strategies for green vehicles, it is essential to take into account consumers' attitudes, beliefs, and behaviors, as well as their WTP premium for these products. This study explores ENK, performance expectations, IO, SN, and PR in relation to BIs and WTP for electric vehicles.

The present research makes three main contributions. Firstly, in contrast to earlier research, this study addresses a gap in the literature by examining all the elements that may influence BI of electric vehicles in China. According to best of authors' knowledge, this is the earliest study to determine the importance of BI and WTP for electric vehicles in the Chinese context. Due to recent economic growth and population growth, the country is facing severe energy-related challenges, necessitating comprehensive research on how electric vehicles can be adopted. Secondly, the Theory of Planned Behavior (TPB) is expanded by incorporating three unique dimensions (performance expectancy, information loaded, and perceived risk) that may influence consumers' BIs and WTP for EVs. Finally, the current study extends the research results in a manner distinct from earlier studies. For example, ENK proved to be a critical dimension in the adoption of electric vehicles. In the same vein, beliefs about the benefits of electric vehicles remain a crucial component of TPB's theoretical framework.

As for the remaining part of the study, it can be summarized as follows: Firstly, the literature regarding the theoretical foundation and hypotheses formulation is reviewed. Secondly, data collection procedures and a sampling structure are explained to determine the methodology for investigating the research questions. Thirdly, we deliberate the results of the empirical analysis. In conclusion, the paper outlines the research limitations, future opportunities for practitioners, and possible policy implications.

## 2. Literature Review and Hypotheses Development

### 2.1. Perceived Environmental Knowledge and Behavioral Intentions

In terms of perceived ENK, it is an individual's knowledge about an ecological system, nature, and the effects of people's actions on the ecosystem [16]. A consumer who holds the concept of ENK is someone who has knowledge of issues related to the environment and knows how to deal with them [17]. Nowadays, information and knowledge play an incredibly crucial role in consumers' decision-making process [18]. The perception of ENK significantly increases customer enthusiasm for buying eco-friendly products. This results in improved consumer behavior as well [19]. It is critical that governing bodies

and companies facilitate consumer acceptance of eco-friendly products through green marketing [20].

In order to improve consumer awareness and knowledge of environmental issues, effective marketing strategies can encourage them to change their lifestyle in favor of products that are environmentally friendly. When consumers receive information and knowledge about a particular product, they are able to determine how they will perceive this product's uniqueness as well as what evidence is used to support their purchase decisions [16,21–23]. As a result, ENK can play a significant role in driving consumer BI [24,25]. A study by [24] examined the influence of perceived ENK, consciousness, and interest on consumer behavior. As a result, we propose the following hypotheses based on the above discussion.

**Hypothesis (H1):** *Perceived ENK will have a positive impact on BI.*

### 2.2. Performance Expectancy and Behavioral Intentions

Literature [24] argue that individuals' expectations, defined as "the likelihood of the outcomes of performing a behavior," may lead to specific behaviors as a result of their BI. The PE of technological systems relates directly to the expectation that the technology will assist the user in performing a routine task, and the perception of usability is influenced by existing constructs [26]. A performance category could include energy savings, cost-savings, reliability, visibility, design, and acceleration in the case of electric vehicles [27,28]). In regard to the use of electric vehicles, performance expectations would cover the expectations of users because of its propensity to provide electricity backup [29]. According to some studies, the reduction in maintenance costs was often a greater motivator for purchasers than the reduction in energy costs. Analyzing a vehicle's BI is necessary for the new technology, including whether the EV's performance will be improved compared with conventional vehicles [29]. Moreover, a significant influence of performance expectations on innovative technology purchases has been found in the literature [30–32]. There was also a significant impact of performance expectations on consumer intentions to adopt cloud technology in Pakistan [33]. In China, [34] found that PE positively impacted consumers' intentions to share electric vehicles. In light of the argument mentioned above, we propose the subsequent hypothesis.

**Hypothesis (H2):** *PE will have a positive impact on BI.*

### 2.3. Information Overloaded and Behavioral Intentions

A variety of disciplines have studied the effects of IO for many years. There is a very straightforward mechanism at work in this construct: as IO overwhelms decision-makers, the amount of information they can process exceeds their capacity, resulting in unsatisfactory decisions. Researchers in traditional retail settings [35,36] have confirmed that the amount of information consumers receive determines their buying behavior.

According to the original TPB, BI is used to predict actual behavior. It has been shown in previous research [37,38] that as the amount of information increases and reaches a maximum, the accuracy and integrity of an intention decrease. It is more important for consumers to have adequate information available to assist them in making decisions instead of having excessive alternative information [29]. When a certain amount of information is processed, IO problems will appear and become more severe [39]. The subjective status of consumers, such as confidence and happiness, reduces the burden of excessive information [40,41]. Muller (1984) found that IO does not systematically affect consumer purchase behavior [29]. It was found by [42] that increases in information quantity are negatively associated with decision accuracy if the quality of the information does not change [29]. Therefore, we suggest the following hypothesis based on the above argument.

**Hypothesis (H3):** *IO will have a negative impact on BI.*

*2.4. Subjective Norms and Behavioral Intentions*

In a nutshell, the SN refers to how consumers interpret their behavior toward the purchase of electric vehicles from their most influential people's perspective [43]. It is a measure of how strongly normative beliefs are held and how motivated people are to adhere to them [44]. The SN refers to an individual's perception that the bulk of his friends and family view his behavior in a particular way [45]. It can also be defined as the psychological pressure to participate in a particular behavior [46]. Previous studies have shown that the more pressure an individual receives from significant individuals, the more likely they are to perform a behavior [47]. According to [48,49], SN is found to positively influence BI. In spite of this, the authors explain that people who perceive a requirement for achieving a particular behavior are more likely to perform that behavior due to greater social pressure [50,51]. According to the Theory of Planned Behavior (TPB), SN positively influences one's intentions to behave in a particular way. The same assertion has been supported by many studies [52,53]. Thus, we suggest the following hypothesis.

**Hypothesis (H4):** *SN will have a positive impact on BI.*

*2.5. Perceived Risk and Behavioral Intentions*

In social psychology research, PR has received much consideration and has been described in a variety of ways. Ref. [54] provide a well-accepted definition of PR as the consumer's expectation of negative utility when purchasing a particular product. It is important to note that PR is a multidimensional variable that includes economic, functional, interpersonal, emotional, behavioral, opportunism, and temporal factors [29].

It is possible that PR influences consumers' purchasing decisions [55]. The results of previous research suggest that consumers' attitudes and intentions to adopt innovative products and services are negatively affected by PR [56–58]. The PR of adopting innovative technology is widely believed to be consumers' most significant obstacle to adoption [59]. These perspectives are also applicable to EVs, since they have been viewed as innovative and revolutionary technologies [60]. According to [61], consumers hesitate to purchase electric vehicles because of safety concerns. An essential factor that may discourage EV acceptance is PR. In light of this, consumers who perceive EV adoption and use risks are more likely to have negative attitudes about these vehicles and reduce their adoption intentions. The risk perception of EVs will cause consumers to doubt whether they are able to improve their travel efficiency, reduce their transportation costs, or provide them with any benefits. Thus, we suggest the following hypothesis.

**Hypothesis (H5):** *PR will have a negative impact on BI.*

*2.6. Behavioral Intentions and Willingness to Pay for EVs*

It has been shown in prior research that consumers are willing to pay more for products that are considered safer or of higher quality [16]. This study conceptualizes WTP more as two independent constructs, despite some research that includes WTP more as part of BI. Firstly, there is no conceptual interchangeability between these two constructs [29]. A buyer's purchase intentions measure their willingness to buy a particular product or service. However, it should be noted that an individual's intention to purchase a product may not necessarily be accompanied by an intention to pay a premium price over alternatives [62]. A consumer's WTP premium price is viewed as a strong indication of loyalty, which is inversely proportional to the magnitude of the premium premium [29]. As the premium gets larger, consumers are less willing to pay it [63]. Therefore, consumer buying intentions may be reduced when the price premium is sufficiently high [64]. Despite their intention to buy, some consumers may not be willing to pay a premium price due to the premium-price

effect. In light of this, this study treats WTP as a premium independently of BI and suggests the following hypothesis (See Figure 1).

**Hypothesis (H6):** *BI will have a positive impact on WTP for EVs.*

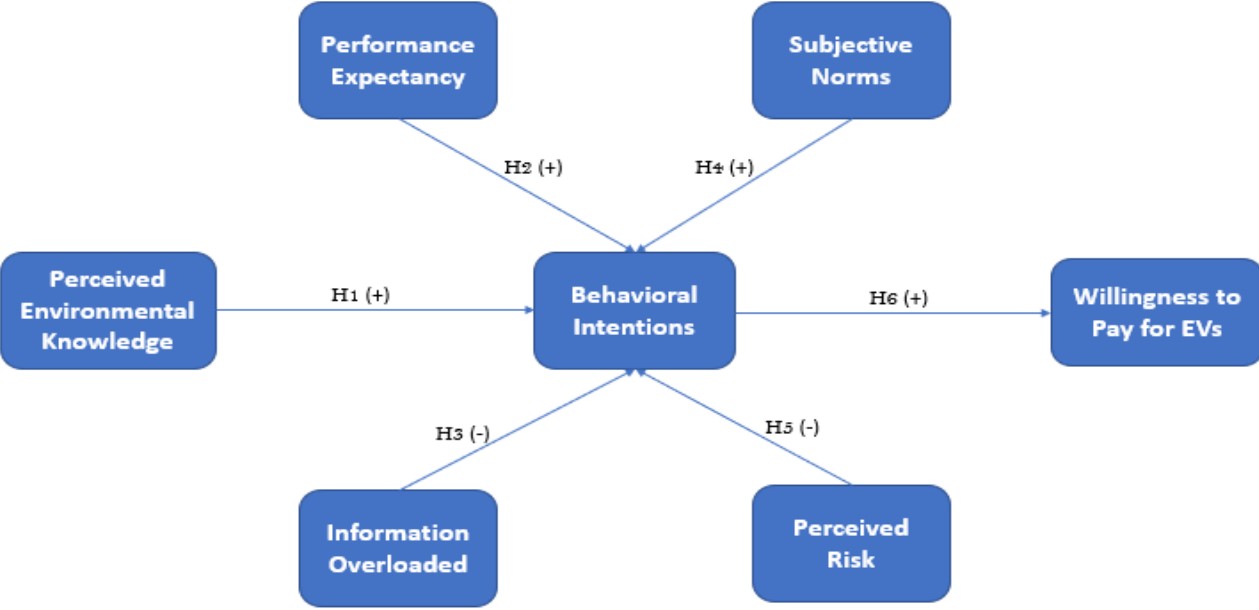

**Figure 1.** Conceptual Framework.

### 3. Method

*3.1. Sample and Data Collection*

In this study, the study's focus was on consumers' BI towards electric vehicles in Beijing, China. The survey area was primarily Beijing because this city was the first to introduce electric vehicles in China, and its sales have always been in the leading position. We conducted the survey between July and August 2022 and distributed questionnaires both online and offline. In order to conduct the survey, "Wenjuanxing," the most popular online survey site in China, was employed. As the study primarily aimed to identify potential electric vehicle buyers in Beijing, we employed convenience sampling technique to select the respondents and sent the questionnaires to professors, fellow students, and friends who work there. An invitation was sent to 350 individuals, and 264 responded online, resulting in a response rate of 75.4%. We eventually obtained 210 valid responses after eliminating 52 questionnaires with similar answers or logic errors, for a valid rate of 79.5%.

During the offline survey setting, the Wudaokou commercial district in Haidian District and the Spring Huimin Auto Show in Beijing were explored, along with the 5th Beijing International Auto Show. It is important to note that the two auto exhibitions chosen in this study are massive exhibitions of automobiles in Beijing. The two auto shows attract a large number of potential auto buyers. The questionnaire was distributed by three university students at the above three locations using a convenience sampling method. In conclusion, 350 survey forms were distributed and gathered in the above three locations. As a result, 288 valid responses were obtained after removing 62 questions with the same answers to most items or with logical errors, resulting in a valid rate of 82.2%. After completing the survey, 498 valid responses were collected. Table 1 provides descriptive statistics for the demographic variables.

**Table 1.** Sample properties.

| Participants' Characteristics | Frequency | Percentage |
|:---:|:---:|:---:|
| *Gender* | | |
| Male | 318 | 63.9 |
| Female | 180 | 36.1 |
| *Age* | | |
| 18–22 | 98 | 19.7 |
| 23–27 | 150 | 30.1 |
| 28–30 | 250 | 50.2 |
| *Education* | | |
| Higher School or below | 10 | 2.0 |
| Intermediate | 52 | 10.4 |
| Bachelors | 240 | 48.2 |
| Masters | 152 | 30.5 |
| PhD or above | 44 | 8.8 |
| *No of cars owned by the household* | | |
| 0 | 40 | 8.0 |
| 1 | 261 | 52.4 |
| >2 | 197 | 39.6 |
| *Household monthly income (CNY)* | | |
| <50,000 | 17 | 3.4 |
| 50,001–100,000 | 234 | 47.0 |
| 100,001–150,000 | 149 | 29.9 |
| 150,001> | 98 | 19.7 |

*3.2. Measures*

We developed a survey questionnaire based on previous studies and customized it to fit the context as shown in Appendix A. A 5-point Likert scale was used to assess questionnaire items adapted from previous literature, (1 = strongly disagree; 5 = strongly agree). A study by [65] provided the basis for evaluating perceived ENK, which was based on five items. A sample item is, "I am very knowledgeable about environmental issues". The six items of PE have been taken from [66] study. A sample item is, "I can learn the EVs usage as a new technology more efficiently." According to [39], we assessed IO by utilizing four items. The following is an example, "There was too much information about EV so I was burdened in handling it". In the [10] study, three items were introduced to measure SN. The following are some examples, "If people around me use electric vehicles, this will prompt me to buy". In [67], four items constitute the PR. Examples of such items include "I worry about whether EVs will really perform as well as traditional gasoline vehicles". We assessed BI using four items from the study of [68]. The following are some examples, "I will try to use the fully automated vehicle if necessary, in life or in work". To evaluate WTP for EVs, we used four items scale from the study of [69]. A sample item is, "I am willing to buy an electric vehicle as I can afford it".

**4. Results**

*4.1. Measurement Model Validation*

Structural equation modeling approach is used to evaluate formulated hypotheses, while SPSS (V 26) and SmartPLS software are used for data analysis purposes. We used correlation analysis to check the interrelationship between variables. After analyzing the test, the results showed a significant correlation between variables (see Table 2). We investigated discriminant validity using the square root of average variance extracted (AVE). The results generated reveal support for discriminant validity because AVE has a higher square root value than its correlation with other constructs [70]. An alternate method to discover discriminant validity is by comparing AVE by MSV value with all variables. If AVE is greater than MSV, discriminant validity is achieved [71]. The square root of the average variance extracted (AVE) is higher than its correlation with other constructs according to discriminant validity estimators [71]. In addition, Table 3 also indicates that all

constructs' composite reliability (CR) is above 0.70, lying between 0.850 to 0.912 [72]. After that, we conducted a convergent validity analysis using AVE and item loadings to check the potential association between these items [73]. Results confirm that the AVE values for every variable are more significant than 0.5, which clears that these variables hit the benchmark and have 50% more variance.

**Table 2.** Discriminant validity.

| Constructs | 1 | 2 | 3 | 4 | 5 | 6 | 7 |
|---|---|---|---|---|---|---|---|
| 1. Behavioral Intention | **0.846** | | | | | | |
| 2. Information Overloaded | 0.719 | **0.779** | | | | | |
| 3. Perceived Environmental Knowledge | 0.831 | 0.737 | **0.821** | | | | |
| 4. Perceived Risk | 0.728 | 0.578 | 0.657 | **0.879** | | | |
| 5. Performance Expectancy | 0.726 | 0.767 | 0.793 | 0.623 | **0.802** | | |
| 6. Social Norms | 0.711 | 0.713 | 0.677 | 0.693 | 0.664 | **0.841** | |
| 7. Willingness to Pay | 0.652 | 0.662 | 0.732 | 0.696 | 0.727 | 0.628 | **0.766** |

The bold values are the $\sqrt{\text{AVE}}$.

**Table 3.** Loading and VIF of the indicators.

| Constructs | Items | Loadings | VIF | $\alpha$ | CR | AVE |
|---|---|---|---|---|---|---|
| **Perceived Environmental Knowledge** | | | | 0.879 | 0.912 | 0.675 |
| | PEK1 | 0.813 | 2.097 | | | |
| | PEK2 | 0.859 | 2.626 | | | |
| | PEK3 | 0.860 | 2.578 | | | |
| | PEK4 | 0.797 | 1.896 | | | |
| | PEK5 | 0.774 | 1.869 | | | |
| **Performance Expectancy** | | | | 0.889 | 0.915 | 0.643 |
| | PE1 | 0.753 | 1.773 | | | |
| | PE2 | 0.825 | 2.442 | | | |
| | PE3 | 0.845 | 2.117 | | | |
| | PE4 | 0.776 | 2.236 | | | |
| | PE5 | 0.789 | 2.071 | | | |
| | PE6 | 0.820 | 2.321 | | | |
| **Information Overloaded** | | | | 0.784 | 0.860 | 0.606 |
| | IO1 | 0.775 | 1.809 | | | |
| | IO2 | 0.794 | 1.923 | | | |
| | IO3 | 0.789 | 1.842 | | | |
| | IO4 | 0.756 | 1.615 | | | |
| **Subjective Norms** | | | | 0.793 | 0.878 | 0.707 |
| | SN1 | 0.883 | 2.319 | | | |
| | SN2 | 0.832 | 2.056 | | | |
| | SN3 | 0.805 | 1.406 | | | |
| **Perceived Risk** | | | | 0.853 | 0.911 | 0.773 |
| | PR1 | 0.895 | 2.344 | | | |
| | PR2 | 0.844 | 1.804 | | | |
| | PR3 | 0.898 | 2.458 | | | |
| **Behavioral Intentions** | | | | 0.867 | 0.910 | 0.716 |
| | BI1 | 0.910 | 2.454 | | | |
| | BI2 | 0.798 | 1.671 | | | |
| | BI3 | 0.812 | 1.959 | | | |
| | BI4 | 0.862 | 2.643 | | | |
| **Willingness to Pay** | | | | 0.768 | 0.850 | 0.587 |
| | WTP1 | 0.825 | 1.568 | | | |
| | WTP2 | 0.711 | 1.345 | | | |
| | WTP3 | 0.773 | 1.506 | | | |
| | WTP4 | 0.751 | 1.602 | | | |

### 4.2. Reliability Analysis

We used the Cronbach-alpha approach to analyze the reliability of all constructs. The results reveal that the Cronbach value for all constructs exceeded the threshold value of 0.70, as recommended by [74], validating the reliability of the data. To examine the coherence of all variables' items, a CR estimation was performed. As a result of the study, it has been determined that the CR values exceed the cutoff value of 0.70 [75]. The results are compiled in Table 3.

### 4.3. Multicollinearity

A regression test is executed to check the multicollinearity issues to find Tolerance and Variance inflation factor (VIF) values. The VIF value should be between 0 and 3 [76]. According to the results (See Table 3), this model does not have any multicollinearity issues because the values of VIF and Tolerance are within the suggested range of each variable and are in line [77].

### 4.4. The Predictive Power of the Model ($Q^2$)

The Stone and Geisser test on SmartPLS were used to assess the predictive utility of our structural model. The predictive power of a conceptual model is determined by its $Q^2$ value being greater than zero ($>0$) for a given conceptual model [78]. Therefore, all of the path model's dependent variables have a $Q^2$ greater than zero, proving that the path model is valid (see Table 4).

**Table 4.** Blindfolding statistics for the general model.

| Construct | SSO | SSE | $Q^2$ (=1-SSE/SSO) |
|---|---|---|---|
| Behavioral Intention | 800 | 635.121 | 0.206 |
| Information Overloaded | 800 | 689.25 | 0.138 |
| Perceived Environmental Knowledge | 800 | 611.58 | 0.235 |
| Perceived Risk | 1000 | 947.225 | 0.052 |
| Performance Expectancy | 800 | 694.772 | 0.132 |
| Subjective Norms | 1000 | 850.359 | 0.150 |
| Willingness to Pay | 1000 | 648.514 | 0.189 |

### 4.5. Structural Model and Hypothesis Outcomes

In addition to testing our hypothesis links with each other and the presented model, we also tested the reliability and validity of our reliable measures. The Value of $R^2$ was found to be 0.766, affirming a meaningful explanation as it achieved the recommended value of 0.35 [79]. Additionally, the covariance-based regression analysis and the SEM algorithm were utilized to test the model relationship. It is clear from the results that the linearity between all links is extreme in terms of the f-value. In addition, we ran various fitness tests to validate that our data match the proposed structural model (i.e., Chi Square = 661.637, NFI = 0.905, and SRMR = 0.051), clearly demonstrating the structural model's fit to our data [80].

In Table 5 and Figure 2, an analysis of the results showed a significant positive impact of perceived ENK on the BI ($H_1-\beta = 0.490$, $p < 0.01$), hence H1 supported the study. Furthermore, PE has a positive and significant association with BI ($H_2-\beta = 0.015$; $p < 0.001$), confirming H2. Similarly, the results reveal that IO has a significant and negative association with BI ($H_3-\beta = -0.125$; $p < 0.001$). Additionally, the direct impact of the fourth hypothesis indicated that SN was positively and significantly related to BI ($H_4-\beta = 0.106$; $p < 0.001$). So, H4 is acknowledged. Moreover, the findings indicate that PR is negatively linked with BI ($H_5-\beta = -0.250$; $p < 0.001$). In last, the direct impact of H6 suggests that BI is positively and significantly related to WTP for EVs ($H_6-\beta = 0.652$; $p < 0.001$).

**Table 5.** Hypotheses testing.

| | Hypotheses | Beta | S.D | *t*-Values | *p*-Values | Decision |
|---|---|---|---|---|---|---|
| H1 | Perceived Environmental knowledge -> Behavioral Intention | 0.490 | 0.085 | 5.743 | 0.000 | Accepted |
| H2 | Performance Expectancy -> Behavioral Intention | 0.015 | 0.077 | 0.198 | 0.013 | Accepted |
| H3 | Information Overloaded -> Behavioral Intention | −0.125 | 0.064 | 1.968 | 0.020 | Accepted |
| H4 | Subjective Norms -> Behavioral Intention | 0.106 | 0.064 | 1.658 | 0.008 | Accepted |
| H5 | Perceived Risk -> Behavioral Intention | −0.250 | 0.073 | 3.444 | 0.001 | Accepted |
| H6 | Behavioral Intention -> Willingness to Pay | 0.652 | 0.040 | 16.399 | 0.000 | Accepted |

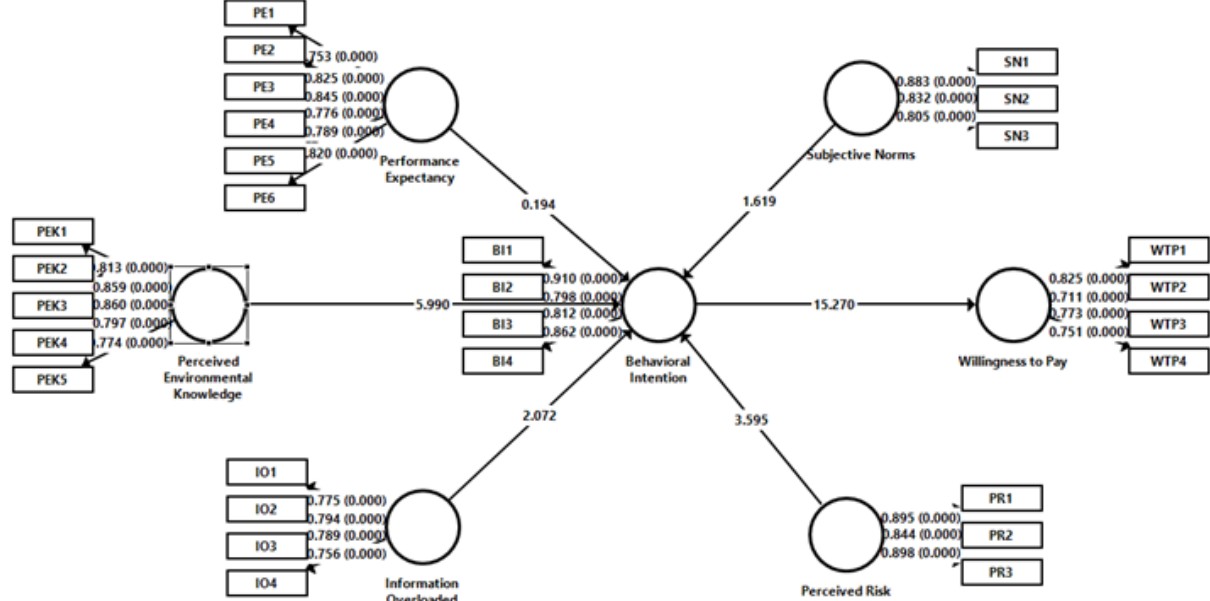

**Figure 2.** Results of Hypotheses.

## 5. Discussion

This study aimed to investigate the factors influencing the BI of individuals from Beijing, China, regarding the purchase of electric vehicles. In general, the findings support the findings of [81] that respondents from Beijing, China are highly concerned about and committed to preserving their environment. In this study, the aim was to investigate and analyze the drivers of consumers' BI in order to better understand their WTP for EVs. In this study, consumers' ENK, PE, IO, SN, and PR are used to estimate consumers' BI of EVs in Beijing, China.

Our study found that all factors positively affect consumer behavior intentions and WTP for EVs. Firstly, the findings indicate that perceived ENK significantly influences consumer behavior toward electric vehicles. Researchers have found that consumers who are knowledgeable about electric cars are more likely to demonstrate positive behavior toward them. As TPB's model shows, ENK is a significant predictor of environmental behavior. These results are consistent with previous research that demonstrated ENK is a reliable predictor for consumer BI [82].

The results of our study also indicate that SN is positively related to EV BI. According to [83], consumers' BI is strongly influenced by SN. It has been shown in previous studies that SN has little effect either on BI or actual behavior [45,83–85]. Additionally, we found that IO is negatively associated with the BI of an electric vehicle. The results are consistent with the research of [39]. In light of these findings, it may be possible to alleviate overload problems by enhancing the quality of EV-related information. Due to this, the quantity and quality of EV-related information play equal roles in determining how much information consumers are overloaded with, depending on the situation.

The results also revealed that SN is a strong influencer of BI. Previous studies have shown that the more pressure an individual receives from significant individuals, the more likely they are to perform a behavior [47]. Inconsistent with the research of [48] and [49], SN is found to positively influence BI. Furthermore, the findings also advocate that PR negatively affects BI. It is possible that PR influences consumers' purchasing decisions [55]. According to [61], consumers have a hesitation to purchase electric vehicles due in part to concerns about safety. An essential factor that may discourage EV acceptance is PR. In light of this, consumers who perceive EV adoption and use risks are more likely to have negative attitudes about these vehicles and reduce their adoption intentions.

Lastly, our findings suggest that BI has strong association with WTP for electric vehicles. In order to determine the WTP of a particular product or service, a buyer must determine their purchasing intentions. However, it should be noted that an individual's intent to buy a product may not necessarily be accompanied by an intention to pay a premium price over alternatives (Gam et al., 2010). According to findings from the study, participants expressed WTP more for an eco-friendly product and their actual purchasing behavior during an auction. It is interesting to note that respondents with high BI indicated that they would be willing to pay 40 percent more than what they actually paid.

## 5.1. Theoretical Implications

As part of the study, theoretical implications are provided that enhance the TPB through its contribution to the existing literature on sustainable consumption. Firstly, this research contains a theoretic framework that includes BI and WTP among consumers and the importance of attributes for electric vehicles. In this study, the emphasis is placed on ENK and attitudes regarding the EVs BI. Environmental development aims to create a positive image of eco-friendly products to promote their purchase [86,87]. Furthermore, the conceptual model presented here is unique in that it has been developed and tested in a developed market (e.g., China). We also found that PR plays an important role in understanding sustainable consumption behavior. In accordance with previous research, a feeling of risk might lead to emotional responses to environmental degradation and its state, which, in turn, may lead to a strong commitment at the individual level to environmental causes. Additionally, cross-national studies have found that perceptions of environmental risks may influence attitudes and behaviors.

## 5.2. Policy Recommendations

Due to our findings, we may need to make some important policy recommendations in order to increase EV buying intent and promote EV development in China. First, the Chinese government and companies need to focus more on improving the infrastructure for fast charging and safety of electric vehicles. This research will significantly impact consumer knowledge and perceptions of electric vehicles. As a result, consumers of electric vehicles may be more inclined to make a purchase.

Second, governments and companies must work together to increase consumers' awareness of electric vehicles. In order to accomplish this, it would be necessary to disseminate information about EVs and demonstrate how they can be used to regenerate the environment while reducing operating costs. The use of exhibitions and experience centers can further educate the public about electric cars. It is possible for consumers to improve their understanding of crucial policy decisions through public service announcements (PSA), e.g., through TV advertisements and expert forums.

Thirdly, it is also important to create and strengthen monetary incentives policies to encourage customers to buy electric cars. In spite of the fact that electric cars (EVs) have become a commercial product in some markets, they are still at the beginning of their development process. A substantial level of support from the government is needed. It is preferable for the government to adopt a five-year tax-free policy on electric vehicles instead of imposing a 1% sales tax. In addition, charging station electricity prices should be lowered. To improve the performance of electric vehicles, incentives should be given

instead of subsidizing electric vehicle sales. It is possible to offer subsidies to non-EV owners so that they can buy EVs.

Finally, it would be helpful if policymakers could provide additional information on the benefits and drawbacks of EVs in order to help people transition to the next stage of evolution. The difference between intrinsic motivations to purchase EVs (and their costs) might also close if they understand their driving habits and the services they can offer. In this way, people will not need incentives for switching to electric vehicles. In spite of this, the effectiveness and efficiency of financial policies are likely to decrease as people shift to electric vehicles. There may be a greater benefit to focusing on behavior modification rather than providing incentives to everyone.

## 6. Conclusions

The results of the study reveal that the level of consumer ENK, PE, IO, SN, and PR regarding electric vehicles is positively and significantly associated with BI. The observational results of the study also show that consumers' BI has a positive and significant influence on WTP for EVs. The study's findings demonstrate that companies must let consumers know about the environmental and safety regulations that require the development of electric vehicles, which validates their higher cost and price. In this study, the importance of ENK and other attributes is highlighted as a factor in purchasing eco-friendly electric vehicles. Finally, the study recommends that future researchers use survey data or research designs to assess how consumers use automated vehicles.

It should also be noted that there are some limitations to this study. There are five factors that significantly promote consumers' EV BI: perceived ENK, PE, IO, SN, and PR measures. Nevertheless, this research only examined the direct effects of the above factors on BI, and the theoretical mechanism analysis is still inadequate. Future studies could explore the indirect effect of the antecedent variables mentioned above on WTP via the mediating role of BI. Additionally, this study was conducted in the city of Beijing, China. However, the findings of future studies should be generalized to more cities based on their electric vehicle usage rate. Moreover, the study applies theory to environmental behavior; major antecedents are included in this study. Future studies can apply eco-conscious models, value-belief-norm theories, and norm activation models to environmental behavior. A future study could include additional psychological variables, such as empathy and morality.

**Author Contributions:** Conceptualization, S.Z.; Data curation, W.G. and S.F.; Formal analysis, Y.Y.; Funding acquisition, H.L. and W.G.; Methodology, J.L.; Writing—original draft, B.L.; Writing—review and editing, H.L. All authors have read and agreed to the published version of the manuscript.

**Funding:** This research was supported by the following funds: The National Social Foundation of China (Grant No. 20BGL247). China Postdoctoral Science Foundation: A study on the mechanism of physician engagement behaviour in online medical communities from the perspective of network effects (No. 2022M710038). Guangxi Science and Technology Base and Talent Special Project: Research on the incentive mechanism of user information sharing in live e-commerce-based on social capital perspective (No. 2020AC19034). 2021 Guangxi 14th Five-Year Education Science Planning Key Special Project: Research on the influence of learning communities on users' online learning behavior in the information technology environment (No. 2021A033). 2021 Guangxi 14th Five-Year Education Science Planning Key Special Project: Research on the influence of short video sharing on Chinese cultural identity of international students in China-taking Jieyin as an example (No. 2021ZJY1607). 2022 Innovation Project of Guangxi Graduate Education: Research on Cultivating Innovation and Practical Ability of Postgraduates in Local Universities in Guangxi. (No. JGY2022122). Guangxi undergraduate teaching reform project in 2022: research on the construction of thinking and government in marketing courses under the online and offline mixed teaching mode. (No. 2022JGB180). Teaching reform project of Guilin University of Electronic Science and Technology: research on the construction of the ideology and politics of the course of Brand Management. (No. JGB202114). Doctoral research initiation project of Guilin University of Electronic Science and Technology: "Research on the incentive mechanism of knowledge sharing in online medical communities" (No. US20001Y).

**Institutional Review Board Statement:** The study was conducted in accordance with the Declaration of Helsinki and approved by the Institutional review board of Guilin University of Electronic Technology (protocol code 894-3, 14-05-2022).

**Informed Consent Statement:** Informed consent was obtained from all subjects involved in the study.

**Data Availability Statement:** The original contributions presented in this study are included in the article. Further inquiries can be directed to the corresponding author.

**Conflicts of Interest:** The authors declare no conflict of interest.

## Appendix A

**Table A1.** Questionnaire.

| Items | Strongly Disagree | 2 | 3 | 4 | Strongly Agree |
|---|---|---|---|---|---|
| **Perceived Environmental knowledge** | | | | | |
| My knowledge of environmental issues is extensive. | | | | | |
| My understanding of environmental issues is greater than the average person. My knowledge of reducing CO2 emissions allows me to choose the least polluting vehicles. | | | | | |
| My understanding of the environmental impacts of vehicle consumption is good. | | | | | |
| My understanding is that hybrid cars are more sustainable than conventional cars. | | | | | |
| **Performance Expectancy** | | | | | |
| My eco-friendly behavior would be enhanced if I used electric vehicles | | | | | |
| My ability to learn the usage of EVs as technological advancement an be improved. | | | | | |
| My fuel and maintenance costs can be reduced by using EVs in comparison to gasoline cars. | | | | | |
| My motivation to buy an electric vehicle is enhanced by the availability of home charging. | | | | | |
| I think there are no disadvantages to using electric vehicles | | | | | |
| My learning and technical activities will be improved if I use electric vehicles | | | | | |
| **Information Overloaded** | | | | | |
| I was burdened with a lot of information about EV. | | | | | |
| I felt that acquiring all the necessary information about EV was difficult due to the abundance of information available. | | | | | |
| In my experience, only a small percentage of the EV information I gathered was useful to me. | | | | | |
| The information I received about EVs was not sufficient to assist me in making a purchasing decision. | | | | | |
| **Subjective norms** | | | | | |
| I will be more likely to purchase an electric vehicle if I see people around me using electric vehicles | | | | | |
| I have been advised to purchase an electric vehicle by people who have influence over me (such as my relatives and friends) | | | | | |
| I will purchase an electric vehicle in response to news media propaganda | | | | | |
| **Perceived Risk** | | | | | |
| I believe that using EVs could involve considerable time losses considering their disadvantages (e.g., limited driving range and long charging times). | | | | | |
| I have concerns regarding the performance of EVs as compared to traditional gasoline powered vehicles | | | | | |
| In my opinion, the environmental crisis has become more serious in recent year. | | | | | |
| **Behavioral Intentions** | | | | | |
| In my personal and professional lives, I wish to use fully electric vehicles whenever possible | | | | | |
| I have a high probability of using a fully electric vehicle in the future | | | | | |
| I will make every effort to utilize a fully electric vehicle if possible | | | | | |
| I am likely to suggest fully electric vehicles to others | | | | | |
| **Willingness to pay for EVs** | | | | | |
| My financial situation permits me to purchase an electric vehicle. | | | | | |
| My preference for electric vehicles is higher than that for gasoline-powered vehicles | | | | | |
| My desire to purchase an electric vehicle is based on its environmental friendliness | | | | | |
| If I do not have cash on hand, I am willing to lease an electric vehicle | | | | | |

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
