# Peer review of "Identifying Intention-Based Factors Influencing Consumers’ Willingness to Pay for Electric Vehicles: A Sustainable Consumption Paradigm"

_sustainability, doi:10.3390/su142416831_

Round 1

Reviewer 1 Report

The study very well and interestingly link consumers’ willingness to pay for electric vehicles
in China and emerging business models. The introduction of the study is very comprehensive, it responds very well to the social demand for solving the problem. I also appreciate the determination of hypotheses based on theoretical frameworks. I also appreciate very well the amount and accuracy of many statistical methods used. The discussion of the results contains the necessary answers to the hypotheses and sufficiently critically compares the obtained results with the results of other authors. 
Please revise the abstract : Research methods and data sources is, in my opinion, insufficient. The data used and especially the methods used should be described in more detail. There is no description of the methods used, Please provide information about what software is used for the analysis, what the analysis procedure will be. In the results, you gradually list a number of methods that should be described and mentioned in this section.

Author Response

The study very well and interestingly link consumers’ willingness to pay for electric vehicles in China and emerging business models. The introduction of the study is very comprehensive, it responds very well to the social demand for solving the problem. I also appreciate the determination of hypotheses based on theoretical frameworks. I also appreciate very well the amount and accuracy of many statistical methods used. The discussion of the results contains the necessary answers to the hypotheses and sufficiently critically compares the obtained results with the results of other authors.

Please revise the abstract: Research methods and data sources is, in my opinion, insufficient. The data used and especially the methods used should be described in more detail. There is no description of the methods used, please provide information about what software is used for the analysis, what the analysis procedure will be. In the results, you gradually list a number of methods that should be described and mentioned in this section.

Authors' response:

Respected reviewer, thank you very much for the positive evaluation of our study. We appreciate your time and dedication in reviewing this submission. Aiming to improve the manuscript, we have followed your suggestions and revised the manuscript.  In this vein, we have revised the abstract. Research methods and data sources have been explained in detail. Software used for data analysis purposes have been mentioned. Discussion of results is also further elaborated for the ease of normal readers.

After addressing all the concerns raised by the worthy reviewer and incorporating all the suggestions, the authors are confident that the paper in its current revised version is improved to the best possible point compared to the previous edition, which will meet the expectations of the respected reviewer. Once again, thank you very much for your professional review of our manuscript. Be safe with your family in the current pandemic period. We look forward to hearing from you soon.

**************************************************************

Reviewer 2 Report

This paper focused on Identifying intention-based factors influencing consumers’ willingness to pay for electric vehicles: A sustainable consumption paradigm. The subject matter of this manuscript fits the journal's scope, and the information included in the manuscript seems not to have been published in any other publication so far. However, it seems difficult to adequately evaluate the value of this study because the explanation of the significance of the study, the description of the interpretation and usefulness of the results obtained by the analysis, and the explanation of the model are insufficient. I would like to ask the authors to consider responding to the following comments:

(1)    Abstract didn’t summaries all the key findings of the manuscript

(2)     Would you explicitly specify the novelty of your work? What progress against the most recent state-of-the-art similar studies was made?

(3)    The Introduction section should be improved. It should be dedicated to presenting a critical analysis of state-of-the-art related work to justify the study's objective. In addition, critical comments should be made on the results of the cited works.

(4)    The main objective of the work must be written in a more precise and concise way at the end of the introduction section.

(5)    Please carefully check recent literature and discuss/cite as you see fit, and update your reference list such as; Prediction of electric vehicle charging duration time using ensemble machine learning algorithm and Shapley additive explanations. A comparative performance of machine learning algorithm to predict electric vehicles energy consumption: A path towards sustainability. Grey wolf optimizer-based machine learning algorithm to predict electric vehicle charging duration time. Electric vehicle energy consumption prediction using stacked generalization: an
ensemble learning approach

(6)    The author is encourage to conduct a further study about how to select the sample size of respondents.

(7)    4. Sampling: although 352 and 498 interviews were collected what was the method of sampling? 

(8)    The distribution of responses are not clear, make a table to more clear the online and offline responses.

(9)    There is a room to improve the research methodology for publishing in an international journal. Furthermore, the numerical experiments were insufficient.

(10)          The reviewer think some figures related to the computation results should be presented to improve the quality of this paper.

(11)          The conclusion section provides a lack of contributions to this manuscript. Provide the key features, merits, and limitations of the proposed approach to clarify the precise boundary of the algorithms. The implication of the proposed method is also required.

(12)          This paper is generally well written, but I found multiple typographic and editorial errors over the entire manuscript, including the equations. The authors need to proofread again carefully.

Author Response

This paper focused on Identifying intention-based factors influencing consumers’ willingness to pay for electric vehicles: A sustainable consumption paradigm. The subject matter of this manuscript fits the journal's scope, and the information included in the manuscript seems not to have been published in any other publication so far. However, it seems difficult to adequately evaluate the value of this study because the explanation of the significance of the study, the description of the interpretation and usefulness of the results obtained by the analysis, and the explanation of the model are insufficient. I would like to ask the authors to consider responding to the following comments:

(1)    Abstract didn’t summaries all the key findings of the manuscript

Authors' response: Respected reviewer, thank you very much for the positive evaluation of our study. We appreciate your time and dedication in reviewing this submission. Aiming to improve the manuscript, we have followed your suggestions and revised the manuscript.  In this vein, we have revised abstract.

 (2)     Would you explicitly specify the novelty of your work? What progress against the most recent state-of-the-art similar studies was made?

Authors' response: We appreciate your concern. Following your suggestion, we have clearly mentioned the novelty of our work. Please refer to the revised manuscript for further details.

(3)    The Introduction section should be improved. It should be dedicated to presenting a critical analysis of state-of-the-art related work to justify the study's objective. In addition, critical comments should be made on the results of the cited works.

Authors' response: Thank you very much for your suggestions. We have thoroughly revised the introduction section and justified the objectives of the study. Please refer to the revised manuscript for further details.

(4)    The main objective of the work must be written in a more precise and concise way at the end of the introduction section.

Authors' response: We appreciate your feedback. In this regard, we have precisely and concisely written the main objective of the work at the end of the introduction section. Please refer to the revised manuscript for further details.

(5)    Please carefully check recent literature and discuss/cite as you see fit, and update your reference list such as; Prediction of electric vehicle charging duration time using ensemble machine learning algorithm and Shapley additive explanations. A comparative performance of machine learning algorithm to predict electric vehicles energy consumption: A path towards sustainability. Grey wolf optimizer-based machine learning algorithm to predict electric vehicle charging duration time. Electric vehicle energy consumption prediction using stacked generalization: an ensemble learning approach

Authors' response: Thank you very much for your suggestions. All the suggested studies have been consulted and cited in the revised manuscript to further improve its contents and quality. Please refer to the revised manuscript for further details.

(6)    The author is encourage to conduct a further study about how to select the sample size of respondents.

Authors' response: We appreciate your feedback. Sample size selection has been explained in detail in the revised manuscript. Please refer to the revised manuscript for further details.

(7)    4. Sampling: although 352 and 498 interviews were collected what was the method of sampling?

Authors' response: Thank you very much for your query. A convenience sampling methodology was used to select the respondents. Please refer to the revised manuscript for further details.

(8)    The distribution of responses are not clear, make a table to more clear the online and offline responses.

Authors' response: We appreciate your feedback. The distribution of the responses has been arranged sequentially. Please refer to the revised manuscript for further details.

(9)    There is a room to improve the research methodology for publishing in an international journal. Furthermore, the numerical experiments were insufficient.

Authors' response: Thank you very much for your suggestions. Methodology has been further improved along with empirical analysis. Please refer to the revised manuscript for further details.

(10) The reviewer think some figures related to the computation results should be presented to improve the quality of this paper.

Authors' response: We appreciate your feedback. Please refer to Figure 2 in this regard.

(11) The conclusion section provides a lack of contributions to this manuscript. Provide the key features, merits, and limitations of the proposed approach to clarify the precise boundary of the algorithms. The implication of the proposed method is also required.

Authors' response: Thank you very much for your suggestions. We have thoroughly improved the conclusion section for better clarity. Please refer to the revised manuscript for further details.

(12) This paper is generally well written, but I found multiple typographic and editorial errors over the entire manuscript, including the equations. The authors need to proofread again carefully.

Authors' response: We appreciate your feedback. We have thoroughly revised the manuscript for language errors. Please refer to the revised manuscript for further details.

After addressing all the concerns raised by the worthy reviewer and incorporating all the suggestions, the authors are confident that the paper in its current revised version is improved to the best possible point compared to the previous edition, which will meet the expectations of the respected reviewer. Once again, thank you very much for your professional review of our manuscript. Be safe with your family in the current pandemic period. We look forward to hearing from you soon.

**************************************************************

Reviewer 3 Report

The manuscript deals with electric vehicles and provides novel insights into the sustainable consumption paradigm. Interestingly, the authors draw on the theory of planned behaviour and expand it by adding the dimensions of performance expectancy, information loaded, and perceived risk. In the empirical part, the researchers adopt a structural equation modeling approach using SmartPLS software. This should be clearly stated at the beginning of the Results section. Except for this minor suggestion for improvement, the manuscript justifies publication.

Author Response

The manuscript deals with electric vehicles and provides novel insights into the sustainable consumption paradigm. Interestingly, the authors draw on the theory of planned behaviour and expand it by adding the dimensions of performance expectancy, information loaded, and perceived risk. In the empirical part, the researchers adopt a structural equation modeling approach using SmartPLS software. This should be clearly stated at the beginning of the Results section. Except for this minor suggestion for improvement, the manuscript justifies publication.

Authors' response:

Respected reviewer, thank you very much for the positive evaluation of our study. We appreciate your time and dedication in reviewing this submission. Aiming to improve the manuscript, we have followed your suggestions and revised the manuscript. In this vein, we have followed your suggestion and revised the manuscript accordingly. Please refer to the revised manuscript for further details.

After addressing all the concerns raised by the worthy reviewer and incorporating all the suggestions, the authors are confident that the paper in its current revised version is improved to the best possible point compared to the previous edition, which will meet the expectations of the respected reviewer. Once again, thank you very much for your professional review of our manuscript. Be safe with your family in the current pandemic period. We look forward to hearing from you soon.

**************************************************************

Round 2

Reviewer 2 Report

The author has addressed all my previous comments adequately